# Smooth Muscle Myosin Localizes at the Leading Edge and Regulates the Redistribution of Actin-regulatory Proteins during Migration

**DOI:** 10.3390/cells11152334

**Published:** 2022-07-29

**Authors:** Ruping Wang, Eylon Arbel, Dale D. Tang

**Affiliations:** Department of Molecular and Cellular Physiology, Albany Medical College, Albany, New York, NY 12208, USA; wangru@amc.edu (R.W.); eylonarb@buffalo.edu (E.A.)

**Keywords:** myosin, leading edge, migration, smooth muscle, actin-associated proteins

## Abstract

Airway smooth muscle cell migration plays an essential role in airway development, repair, and remodeling. Smooth muscle myosin II has been traditionally thought to localize in the cytoplasm solely and regulates cell migration by affecting stress fiber formation and focal adhesion assembly. In this study, we unexpectedly found that 20-kDa myosin light chain (MLC_20_) and myosin-11 (MYH11), important components of smooth muscle myosin, were present at the edge of lamellipodia. The knockdown of MLC_20_ or MYH11 attenuated the recruitment of c-Abl, cortactinProfilin-1 (Pfn-1), and Abi1 to the cell edge. Moreover, myosin light chain kinase (MLCK) colocalized with integrin β1 at the tip of protrusion. The inhibition of MLCK attenuated the recruitment of c-Abl, cortactin, Pfn-1, and Abi1 to the cell edge. Furthermore, MLCK localization at the leading edge was reduced by integrin β1 knockdown. Taken together, our results demonstrate that smooth muscle myosin localizes at the leading edge and orchestrates the recruitment of actin-regulatory proteins to the tip of lamellipodia. Mechanistically, integrin β1 recruits MLCK to the leading edge, which catalyzes MLC_20_ phosphorylation. Activated myosin regulates the recruitment of actin-regulatory proteins to the leading edge, and promotes lamellipodial formation and migration.

## 1. Introduction

Identifying the subcellular localization of a protein is fundamental for understanding its function. Myosin II is a hexameric protein comprised of two myosin heavy chains, two 20-kDa myosin light chains (MLC_20_), and two 17-kDa essential myosin light chains (MLC_17_) [1]. Smooth muscle myosin II has been traditionally thought to localize in the cytoplasm solely [1]. The phosphorylation of MLC_20_ by myosin light chain kinase (MLCK) initiates contractile filament sliding and smooth muscle contraction [2]. In addition, MLC_20_ phosphorylation modulates cell migration by affecting stress fiber formation, focal adhesion assembly, and the retraction of the rear [3,4]. In breast cancer cells, non-muscle myosin IIA localizes at the leading edge, facilitating lamellipodia formation, focal adhesion turnover, and actin polymerization [5]. However, it is currently unknown whether smooth muscle myosin localizes in other structures of the cell. This question is fundamental because studies on intracellular localizations of myosin may reveal its new functions in smooth muscle, an important cell type affecting respiratory and cardiovascular functions.

Airway smooth muscle cell migration plays an essential role in airway development, repair, and remodeling. In the initial stage of migration, cells explore their environment by forming several small protrusions. Once they decide on a moving direction, cells form a large protrusion called lamellipodia at the leading cell edge along the migration path, which is critical for directional migration [6,7,8]. The formation of lamellipodia is largely driven by the recruitment of actin-regulatory proteins to the leading edge, which promotes local actin filament branching/polymerization and extends the membrane forward [6,9]. For instance, c-Abl (Abelson tyrosine kinase, Abl) is recruited to the leading edge, which catalyzes cortactin phosphorylation at Tyr-421, and promotes the recruitment of profilin-1 (Pfn-1) to the leading cell edge and transports G-actin to the barbed end of actin filaments [8,10]. Cortactin also activates N-WASP (Neuronal Wiskott–Aldrich Syndrome Protein) and enhances local actin branching and network formation [8,10]. In addition, c-Abl activates Abi1 (Abl interactor-1) by directly interacting with Abi1 [10,11,12]. Abi1 promotes actin cytoskeletal reorganization by modulating nucleation-promoting factors such as WAVE (WAS family member) and N-WASP, which activate Arp2/3 (Actin Related Protein 2/3 complex)-dependent actin filament branching and assembly [11,13,14]. Despite the importance, the mechanisms that regulate lamellipodial formation remain unclear.

Similar to myosin, MLCK has been thought to localize with myosin in the cytoplasm and catalyze MLC_20_ phosphorylation [2,15], which facilitates cell migration [3,4]. In addition, MLCK is believed to interact with F-actin, and affects membrane tension and the migration of mouse intestinal smooth muscle cells [16]. The role of MLCK in protein recruitment during migration has been under-investigated.

Moreover, β-integrins couple with α-integrins to form the transmembrane adhesion receptors that connect the actin cytoskeleton to the extracellular matrix. β-integrins have been shown to position at the leading cell edge and facilitate the recruitment of actin-associated proteins and cell migration [8,9,10,17].

In this study, we unexpectedly find that a fraction of myosin localizes at the edge of lamellipodia, which regulates the recruitment of actin-regulatory proteins and cell migration. Integrin β1 controls the distribution of MLCK and myosin activation to the leading edge.

## 2. Materials and Methods

### 2.1. Cell Culture

Human airway smooth muscle (HASM) cells were prepared from human bronchi and adjacent tracheas obtained from the International Institute for Advanced Medicine [11,18,19,20]. Human tissues were non-transplantable and informed consent was obtained from all subjects for research. This study was approved by the Albany Medical College Committee on Research Involving Human Subjects. All methods were performed in accordance with the relevant guidelines and regulations of the Albany Medical College Committee on Research Involving Human Subjects. Briefly, muscle tissues were incubated for 20 min with dissociation solution (130 mM NaCl, 5 mM KCl, 1.0 mM CaCl_2_, 1.0 mM MgCl_2_, 10 mM Hepes, 0.25 mM EDTA, 10 mM D-glucose, 10 mM taurine, pH 7, 4.5 mg collagenase (type I), 10 mg papain (type IV), 1 mg/mL BSA and 1 mM dithiothreitol). All enzymes were purchased from Sigma-Aldrich. The tissues were then washed with Hepes-buffered saline solution (composition in mM: 10 Hepes, 130 NaCl, 5 KCl, 10 glucose, 1 CaCl_2_, 1 MgCl_2_, 0.25 EDTA, 10 taurine, pH 7). The cell suspension was mixed with Ham’s F12 medium supplemented with 10% (*v*/*v*) fetal bovine serum (FBS) and antibiotics (100 units/mL penicillin, 100 µg/mL streptomycin). Cells were cultured at 37 °C in the presence of 5% CO_2_ in the same medium. The medium was changed every 3–4 days until cells reached confluence, and confluent cells were passaged with trypsin/EDTA solution [21]. Cells from three donors were used for the experiments. In some cases, duplicated experiments were performed for cells from a donor [22,23].

### 2.2. Wound Healing Assay

An artificial wound was made in the monolayer of cells cultured in a 6-well plate by scraping a 10 μL pipette tip across the bottom of the plate. Cells in the medium containing 10% FBS were allowed to migrate for 12 h in a 37 °C incubation chamber with 5% CO_2_. Cell images were taken using a Leica DMI600 microscope system. The remaining open area of the wound was measured using the NIH ImageJ software.

### 2.3. Immunoblot Analysis and Coimmunoprecipitation

The immunoblot analysis of cell lysis and coimmunprecipitation were performed using the experimental procedures previously described [18,22,24,25,26,27,28]. Briefly, cells were lysed in SDS sample buffer composed of 1.5% dithiothreitol, 2% SDS, 80 mM Tris-HCl (pH 6.8), 10% glycerol, and 0.01% bromophenol blue. The lysates were boiled in the buffer for 5 min and separated by SDS-PAGE. Proteins were transferred to nitrocellulose membranes. The membranes were blocked with bovine serum albumin or milk for 1 h and probed with the use of primary antibodies followed by horseradish peroxidase-conjugated secondary antibodies (ThermoFisher Scientific, Waltham, MA, USA). Proteins were visualized by enhanced chemiluminescence (ThermoFisher Scientific) using the GE Amersham Imager 600 system. For coimmunoprecipitation, precleared cell extracts were incubated overnight with corresponding antibodies and then incubated for 3 h with 150 µL of protein A/G PLUS agarose beads (Santa Cruz Biotechnology, Dallas, TX, USA). Immunocomplexes were washed four times in buffer containing 50 mM Tris-HCl (pH 7.6), 150 mM NaCl and 0.1% Triton X-100. The immunoprecipitates were separated by SDS-PAGE followed by transfer to nitrocellulose membranes. The membranes of immunoprecipitates were probed with the use of corresponding antibodies.

MLC_20_ antibody was custom-made as previously described [29]. Myosin heavy chain (MYH11) antibody was purchased from Santa Cruz Biotechnology (SC-6956, L/N C1815). Glyceraldehyde 3-phosphate dehydrogenase (GAPDH) antibody was acquired from Santa Cruz Biotechnology (#SC-32233, K0315 and Ambien (#AM4300, L/N 1311029). Integrin β1 antibody (1:1000) was purchased from Santa Cruz Biotechnology (SC-374429, K1620). Antibodies against MLC_20_, integrin β1, and MYH11 were validated by KD experiments. GAPDH antibody was validated by examining the molecular weight of detected bands. Finally, vendors have provided a datasheet to show that antibodies were validated by positive controls. The levels of proteins were quantified by scanning the densitometry of immunoblots (Fuji Multi Gauge Software or GE IQTL software, Boston, MA, USA). The luminescent signals from all immunoblots were within the linear range.

### 2.4. Cell KD and Cell Transfection 

For MLC_20_ and integrin β1 KD, control siRNA (SC-37007), MLC_20_ siRNA (SC-45414), and integrin β1 siRNA (SC-35674) were purchased from Santa Cruz Biotechnology. HASM cells were transfected with siRNA according to the manual of the manufacturer (Santa Cruz Biotechnology, Dallas, TX, USA). For MYH11 KD, control construct (sc-418922) and MYH11 CRISPR/Casp KO plasmids (sc-400695) were purchased from Santa Cruz Biotechnology. The experiments were performed according to the manual of the manufacturer.

### 2.5. Fluorescent Microscopy

Cells were plated in dishes containing collagen-coated coverslips and cultured in a CO_2_ incubator for 30 min, followed by fixation and permeabilization [21,30,31]. These cells were immunofluorescently stained using a primary antibody followed by an appropriate secondary antibody conjugated to Alexa-488 or Alexa-555 (Invitrogen, ThermoFisher, Waltham, MA, USA). For the visualization of F-actin, cells were stained with rhodamine-phalloidin. The cellular localization of fluorescently labeled proteins was viewed under a high-resolution digital fluorescent microscope (Leica DMI600) or a Leica SPE confocal microscope. For quantitative analysis, the LAS X software (Leica) was used to perform at least 5 line scans for each cell. The average intensity of at least 5 line scans across each cell was used for analysis.

c-Abl antibody was purchased from Cell Signaling (#2862S, L/N 15) and was validated by using corresponding KD cells [8]. Abi1 antibody was purchased from Sigma (#A5106–200UL, L/N 076M4842V) and validated by using corresponding KD cells [27]. Cortactin antibody was purchased from Santa Cruz Biotechnology (#SC-11408/LN F3010 and #SC-55578/LN Z0417) and validated by using cortactin KD cells [19]. Pfn-1 antibody was purchased from Sigma and Santa Cruz Biotechnology (Sigma, #p7624, L/N 015m4753v; Santa Cruz, #sc-166191, L/N 1-11409). pMLC_20_ (S19) antibody was purchased from Cell Signaling (#3675S, 3/5/6). MLCK antibody was purchased from Invitrogen (#MA5-15176, LN# VF3018495). Antibodies against Pfn-1, pMLC_20_, and MLCK were validated by examining the molecular weight of the detected bands.

### 2.6. Statistical Analysis

All statistical analyses were performed using Prism software (GraphPad Software, San Diego, CA, USA). Differences between pairs of groups were analyzed by Student’s *t*-test. A comparison among multiple groups was performed by one-way or two-way ANOVA followed by a post hoc test (Tukey’s multiple comparisons). Values of n refer to the number of experiments used to obtain each value. *p* < 0.05 was considered to be significant [26,27,32].

## 3. Results

### 3.1. Smooth Muscle Myosin Localizes at the Leading Cell Edge

To assess the spatial localization of myosin during spreading (an early event of cell migration), we plated human airway smooth muscle (HASM) cells onto collagen-coated coverslips for 30 min and fluorescently stained them for MLC_20_ and F-actin. We noticed that MLC_20_ localized in the cytoplasm containing stress fibers. Unexpectedly, a relatively high intensity of MLC_20_ was found at the leading edge, which was colocalized with F-actin (Figure 1A). Moreover, myosin-11 (MYH11, smooth muscle-specific myosin heavy chain) [33] was also at the tip of the leading cell edge, which was colocalized with MLC_20_ (Figure 1B). Because myosin activation by MLC_20_ phosphorylation is associated with migration [3,4], we also assessed the spatial localization of phosphorylated MLC_20_, which was found at the leading edge (Figure 1C). These results suggest that a fraction of smooth muscle myosin localizes at the leading cell edge during migration.

### 3.2. Knockdown of MLC_20_ and MYH11 Inhibits Smooth Muscle Cell Motility

To assess the role of myosin in cell migration, we used siRNA to knock down MLC_20_ in smooth muscle cells, which was verified by immunoblot analysis (Figure 2A). Because the complete knockdown (KD) of MLC_20_ impairs smooth muscle cell viability, we used the experimental condition in which MLC_20_ expression was downregulated by approximately 50%, for the following experiments. We used the wound healing assay to evaluate cellular migration. The KD of MLC_20_ attenuated the directional motility of smooth muscle cells (Figure 2B). Moreover, we used the Crispr/Cas9 technology to knock down MYH11 in smooth muscle cells (Figure 2C). The downregulation of MYH11 also reduced cell migration (Figure 2D). We also noticed that approximately 50% KD of MLC_20_ or MYH11 KD attenuated cell migration by 60%. These findings indicate that HASM cell migration is sensitive to myosin KD.

### 3.3. KD of MLC_20_ Reduces Recruitment of c-Abl, Cortactin, Pfn-1, and Abi1 to the Cell Edge

Because the actin-regulatory protein c-Abl is recruited to the leading cell edge and plays an important role in regulating smooth muscle cell migration [8], we evaluated the effects of MLC_20_ KD on the cellular localization of c-Abl during spreading. In control cells, c-Abl was found at the cell edge; however, MLC_20_ KD cells did not have a well-defined leading edge. More importantly, the intensity of c-Abl was reduced at the cell periphery (Figure 3A). Furthermore, we also assessed the effects of MLC_20_ KD on the spatial localization of cortactin, Pfn-1, and Abi1 because these proteins participate in the protrusion formation of migratory cells [9,27,34,35]. The KD of MLC_20_ attenuated the localization of cortactin, Pfn-1, and Abi1 at the cell periphery (Figure 3A). In addition, MLC_20_ KD reduced the F-actin staining intensity at the cell edge (Figure 3A). Quantitative analysis showed that MLC_20_ KD significantly diminished the positioning of these proteins at the cell edge (Figure 3B,C).

### 3.4. KD of MYH11 Reduces Recruitment of c-Abl, Cortactin, Pfn-1, and Abi1 to the Cell Edge

To further explore the role of myosin, we knocked down MYH11 expression and assessed the spatial distribution of these proteins. The knockdown of MYH11 reduced the localization of these actin-regulatory proteins at the cell edge (Figure 4). Together with the MLC_20_ KD experiment, these findings demonstrate that myosin plays an important role in facilitating the recruitment of actin-regulatory proteins to the cell edge.

### 3.5. MLCK and Integrin β1 Colocalizes at the Leading Edge

Because phosphorylated MLC_20_ positioned at the leading edge (Figure 1C), we sought to assess the spatial distribution of its upstream regulator MLCK. MLCK was positioned at the tip of lamellipodia (Figure 5A). In addition, integrin β1 is highly expressed in smooth muscle cells and recruits migration-associated proteins to the cell edge [8,36]. This raises the possibility that integrin β1 may colocalize with MLCK. Confocal microscopy showed that integrin β1 and MLCK colocalized at the tip of cell protrusion (Figure 5A). Furthermore, we used the coimmunoprecipitation assay to evaluate the interaction of integrin β1 with MLCK. Integrin β1 was found in MLCK immunoprecipitates (Appendix A).

### 3.6. Inhibition of MLC_20_ Phosphorylation Reduces the Recruitment of Actin-regulatory Proteins

Next, we tested whether the inhibition of MLC_20_ phosphorylation by ML7 (an MLCK inhibitor) affects the physical localization of actin-associated proteins. Treatment with ML7 reduced the localization of c-Ab, cortactin, Pfn-1, Abi1, and F-actin at the cell edge (Figure 5B,C). These results indicate that MLC_20_ phosphorylation by MLCK plays a role in regulating the positioning of actin-regulatory proteins at the tip of the cell front.

### 3.7. Inhibition of Actin Polymerization Affects the Recruitment of Actin-regulatory Proteins

MLC_20_ phosphorylation regulates myosin filament assembly and affects actin network remodeling; moreover, actin filament formation regulates the trafficking of β-catenin and GLUT 4 [18,37]. Thus, we hypothesized that F-actin may affect the distribution of actin-regulatory proteins. To test this, we treated cells with the actin polymerization inhibitor latrunculin A and assessed the spatial distribution of these proteins. Because high-dose latrunculin A completely blocks cell adhesion, we treated cells with low-dose (10 nM) latrunculin A, which effectively inhibited F-actin staining and stress fiber formation (Figure 5B). The cells treated with latrunculin A did not display a well-defined leading edge but had several “spikes” at the cell periphery. The results are consistent with the concept that actin polymerization is important for lamellipodial formation. More importantly, c-Abl, Pfn-1, and Abi1 were not localized at the cell edge (Figure 5B,C). Interestingly, a fraction of cortactin was found in a few spikes. However, cortactin did not localize at the large portion of cell edges (Figure 5B,C).

*Integrin β1 controls the localization of MLCK at the cell edge.* Since we found that MLCK and integrin β1 co-localized at the tip of the cell protrusion (Figure 5A), we hypothesized that integrin β1 may affect the localization of MLCK. To test this, we knocked down integrin β1 in the cells (Figure 6A) and discovered that integrin β1 KD reduced the localization of MLCK and its substrate pMLC_20_ at the cell edge (Figure 6B).

## 4. Discussion

Smooth muscle myosin II has been traditionally thought to localize in the cytoplasm solely [1] and modulates cell migration by affecting stress fiber formation, focal adhesion assembly, and the retraction of the rear [3,4]. In this study, MLC_20_ and MYH11 were found at the edge of lamellipodia. Moreover, phosphorylated MLC_20_ was also found at the cell edge. The results demonstrate that smooth muscle myosin, particularly activated myosin, localizes at the leading edge beside the cytoplasm. To the best of our knowledge, this is the first evidence to suggest that smooth muscle myosin localizes at the leading cell edge.

It is well established that the recruitment of actin-regulatory proteins to the leading edge is critical for regional actin network formation and lamellipodial formation. c-Abl, cortactin, Pfn-1, and Abi1 localize at the leading cell edge, as evidenced by immunostaining and/or tagged-protein live cell imaging; the disruption of their positioning at the cell edge by molecular manipulation inhibits migration [8,9,27,38]. The localization of myosin at the leading edge prompts us to investigate the role of myosin in the redistribution of actin-associated proteins. The KD of MLC_20_ or MYH11 is sufficient to attenuate the recruitment of c-Abl, cortactin, Pfn-1, and Abi1 to the cell edge. In addition, the KD of MLC_20_ or MYH11 reduced lamellipodial formation and migration. These findings demonstrate that myosin is important for the redistribution of actin-regulatory proteins to the leading edge, protrusion formation, and migration. Myosin largely interacts with actin filaments to form stress fiber in cultured cells. The colocalization of myosin with F-actin at the cell edge supports this concept. We do not exactly know how myosin KD affects the translocation of these proteins. Non-muscle myosin II has been shown to regulate the translocation of the adhesion-associated proteins of differentiated and non-motile smooth muscle cells [39]. It is possible that the lack of smooth muscle myosin may disrupt the organization of myosin filaments in cells, which inhibits the translocation of these proteins. It is also possible that the disarrangement of myosin filaments by myosin deficiency may affect the actin network, which subsequently influences the recruitment of these proteins to the leading edge.

In this study, pMLC_20_ is found at the leading cell edge. MLC_20_ phosphorylation is known to regulate myosin filament assembly and the sliding of contractile filaments, which can break actin filaments [40,41] facilitating actin depolymerization, which generates more actin monomers for the next round of polymerization [10,32]. Thus, we speculate that the phosphorylation of MLC_20_ may promote actomyosin network reorganization at the leading cell periphery.

MLCK is a major player that catalyzes MLC_20_ phosphorylation in smooth muscle [42]. In this report, we notice that MLCK is found at the cell edge. The inhibition of MLCK reduces the recruitment of actin-regulatory proteins. Unlike other protein kinases, MLCK specifically catalyzes the phosphorylation of MLC_20_ [42]. Thus, it is likely that the inhibition of MLCK reduces MLC_20_ phosphorylation, which subsequently affects the reorganization of the actomyosin network and the recruitment of these proteins to the tip of lamellipodia. In addition, it is also possible that MLCK at the leading edge catalyzes MLC_20_ phosphorylation in the cortex, which subsequently diffuses in the cytoplasm and promotes stress fiber formation. Future studies are required to test this possibility.

We also observed that inhibition of actin polymerization reduces the redistribution of c-Abl, cortactin, Pfn-1, and Abi1. Actin filament dynamics have been shown to regulate the trafficking of β-catenin and GLUT 4 [18,37]. It is possible that actin polymerization may create a local environment for actin-regulatory proteins to have better access to the cytoplasmic domain of integrin-β or other protein complexes [18,43]. Moreover, cells treated with low-dose latrunculin A display a “spike” instead of a well-defined protrusion, suggesting a critical role for actin polymerization in protrusion formation. We also notice that a small pool of cortactin is localized at the “spikes”. This may be because actin polymerization is not completely blocked or cortactin is less sensitive to actin disassembly.

MLCK facilitates stress fiber formation by regulating the phosphorylation of MLC_20_ [44]. In this study, MLCK is colocalized with integrin β1 at the tip of lamellipodia. The KD of integrin β1 reduces the localization of MLCK and pMLC_20_. The results suggest that integrin β1 is important for the recruitment of MLCK and pMLC_20_. Currently, we do not know how MLCK interacts with integrin β1. One possibility is that MLCK may associate with integrin β1 via other proteins such as cortactin [45]. It is also possible that MLCK may directly interact with the cytoplasmic domain of integrin β1. Future studies are required to test the possibilities.

Conclusion: We provided the first evidence that smooth muscle myosin is found at the tip of lamellipodia, which is important for the recruitment of actin-regulatory proteins and F-actin formation. Furthermore, integrin β1 recruits MLCK to the edge of lamellipodia, which catalyzes MLC_20_ phosphorylation. Activated myosin regulates the recruitment of actin-regulatory proteins to the leading edge, and promotes lamellipodial formation and migration (Figure 7).

## Figures and Tables

**Figure 1 cells-11-02334-f001:**
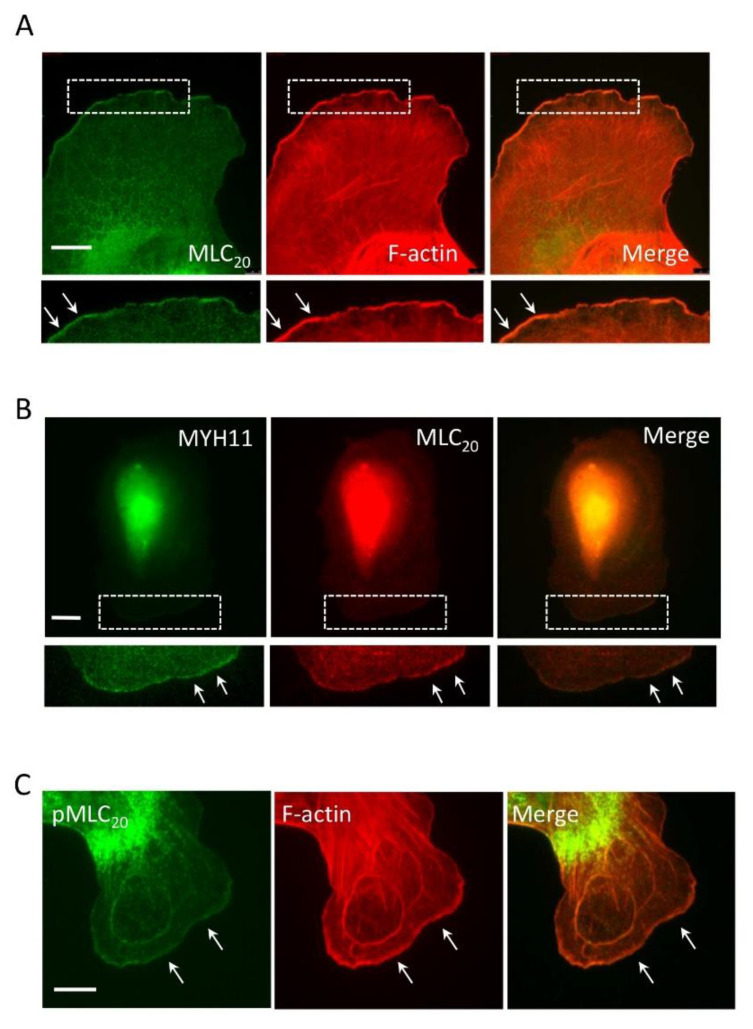
**Smooth muscle myosin localizes at the tip of lamellipodia.** (**A**) Human airway smooth muscle (HASM) cells were plated onto collagen-coated coverslips for 30 min and stained for 20-kDa myosin light chain (MLC_20_) and F-actin. MLC_20_ is found at the leading cell edge. In addition, F-actin is also localized at the cell edge. (**B**) Myosin-11 (MYH11) colocalizes with MLC_20_ at the leading edge. (**C**) Phosphorylated MLC_20_ (pMLC_20_) is found to position at the leading edge. The arrows point to the leading edge. Scale bar: 10 µm.

**Figure 2 cells-11-02334-f002:**
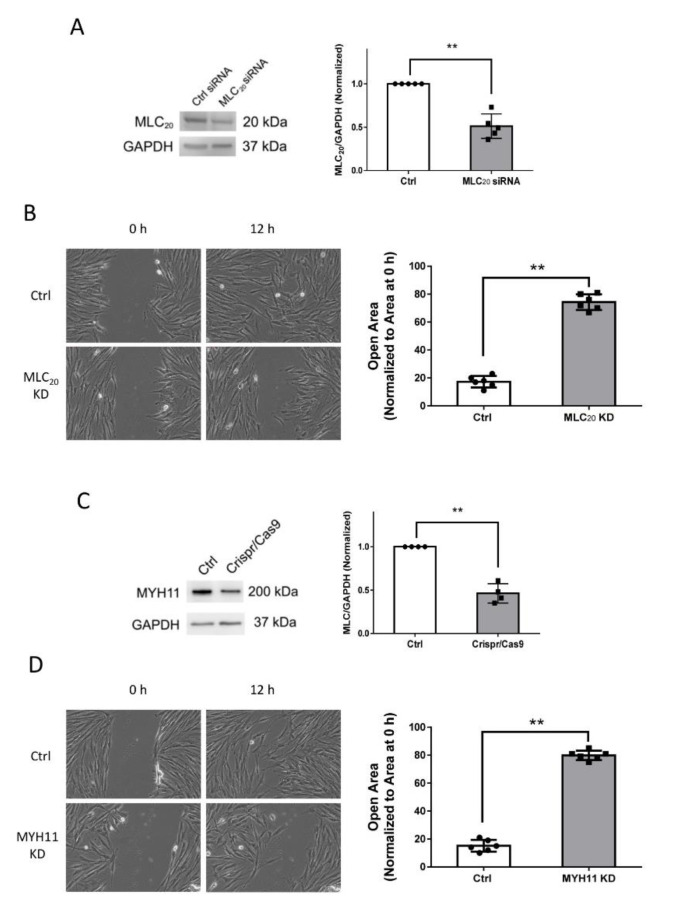
**Knockdown (KD) of MLC_20_ or MYH11 attenuates cell migration.** (**A**) Human airway smooth muscle (HASM) cells treated with control (Ctrl) siRNA or MLC_20_ siRNA were evaluated by immunoblot analysis. MLC siRNA reduces the expression of MLC_20_ in HASM cells. Data are mean values of experiments from five cultures from three donors. Error bars indicate SD. (**B**) The migration of HASM cells was examined by using the wound healing assay. *n* = six experiments from three donors. Error bars indicate SD. (**C**) Cells were treated with Ctrl or Crispr/Cas9 constructs followed by immunoblot analysis. Data are the mean values of experiments from four cultures from three donors. Error bars indicate SD. (**D**) MYH11 KD reduces the migration of HASM cells. n = six experiments from three donors. Error bars indicate SD. ** *p* < 0.01. Student’s *t*-test was used for statistical analysis.

**Figure 3 cells-11-02334-f003:**
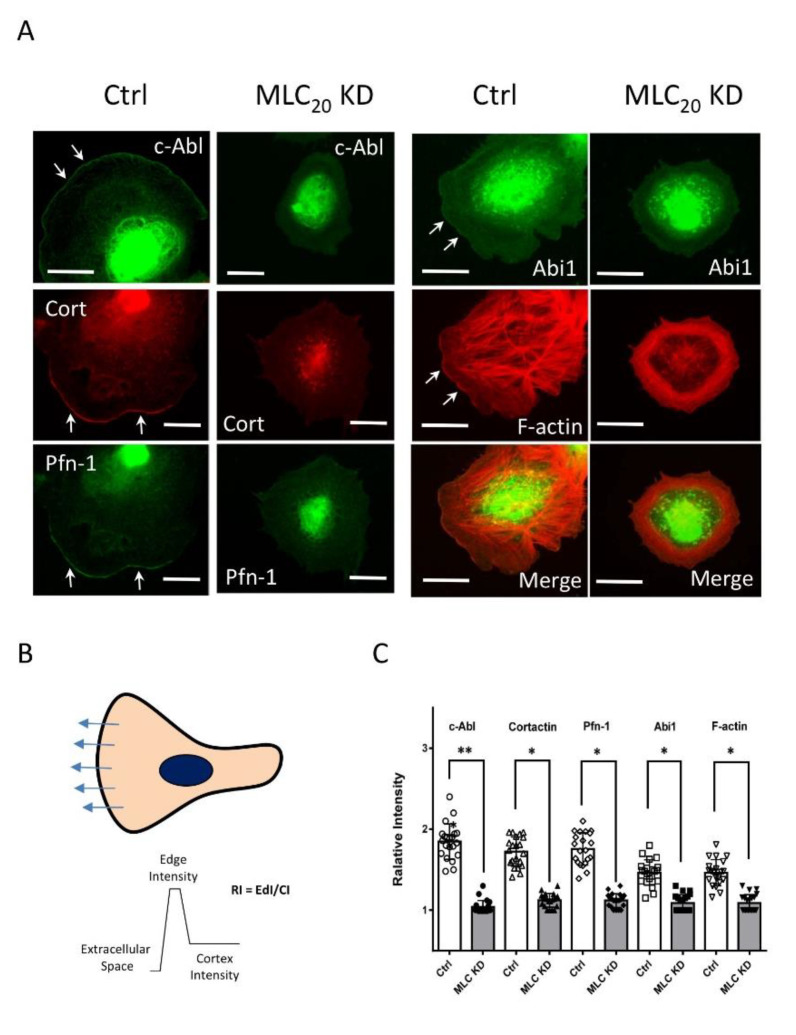
**MLC_20_ regulates the recruitment of c-Ab, cortactin, Pfn-1, and Abi1 to the cell edge.** (**A**) Ctrl and MLC_20_ KD cells were plated onto collagen-coated coverslips for 30 min followed by immunofluorescence and fluorescence analysis. MLC_20_ KD attenuates the localization of c-Abl, cortactin (cort), Pfn-1, Abi1, and F-actin at the cell edge. The arrows point to the leading edge. Scale bar: 10 µm. (**B**) Illustration of quantitative analysis. An average of at least 5 line scans across each cell is used for analysis. Relative Intensity (RI) = Edge Intensity (EdI)/Cortex Intensity (CI). (**C**) Data are mean values of experiments from at least 20 cells for each group. Error bars indicate SD. One-way ANOVA was used for statistical analysis. ** *p* < 0.01; * *p* < 0.05.

**Figure 4 cells-11-02334-f004:**
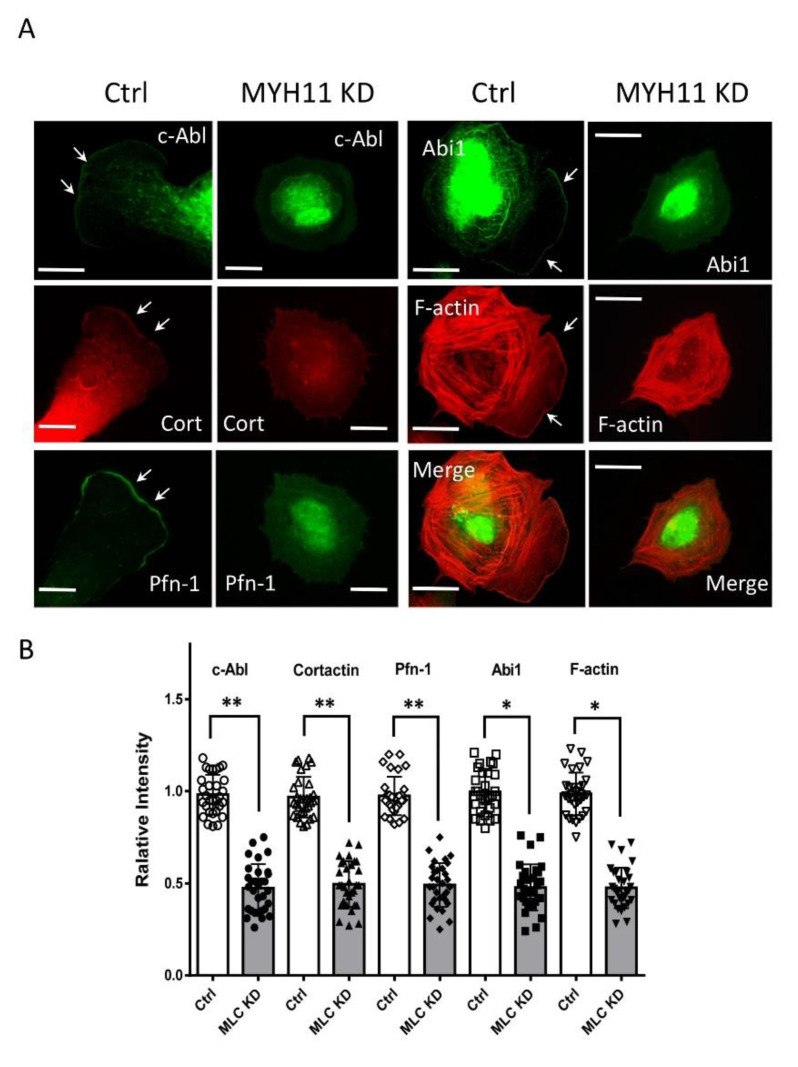
**MYH11 orchestrates the positioning of c-Ab, cortactin, Pfn-1, and Abi1 to the cell edge.** (**A**) Ctrl and MYH11 KD cells were plated onto collagen-coated coverslips for 30 min followed by immunofluorescence and fluorescence analysis. The arrows point to the leading edge. Scale bar: 10 µm. (**B**) Data are mean values of experiments from at least 20 cells for each group. Error bars indicate SD. One-way ANOVA was used for statistical analysis. ** *p* < 0.01; * *p* < 0.05.

**Figure 5 cells-11-02334-f005:**
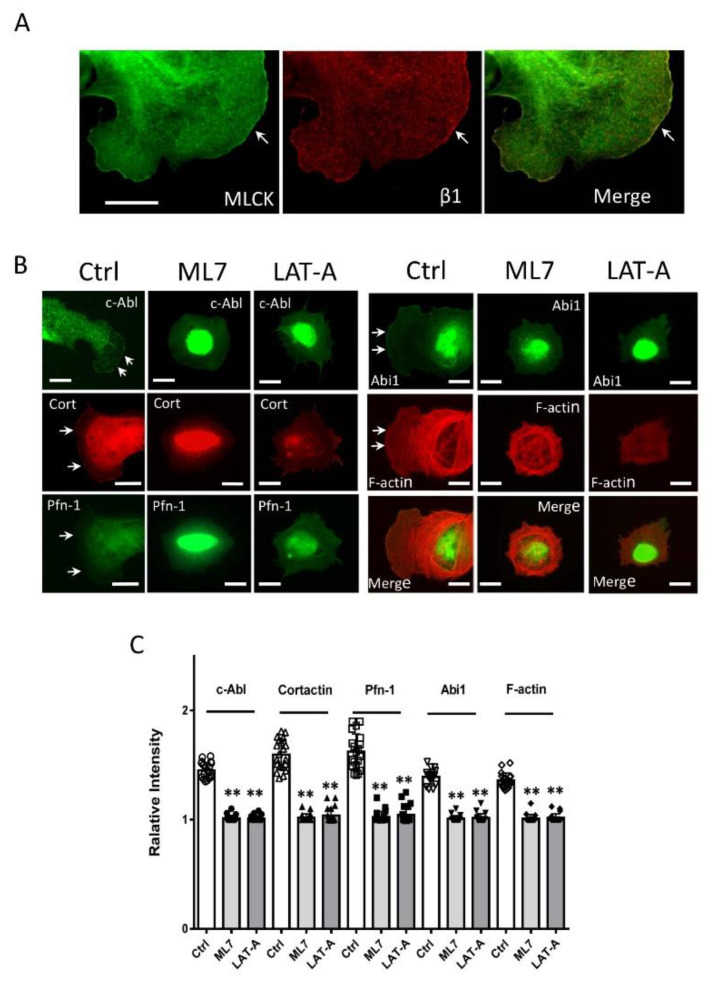
**MLC_20_ phosphorylation and actin polymerization regulate the recruitment of actin-regulatory proteins to the cell edge.** (**A**) Myosin light chain kinase (MLCK) colocalizes with integrin β1 at the cell edge. Cells were immunostained for MLCK and integrin β1. The arrows point to the leading edge. Scale bar: 10 µm. (**B**) Cells were plated onto collagen-coated coverslips in the absence or presence of 1 µM ML7 or 10 nM latrunculin A (LAT-A) for 30 min. They were then stained for the indicated proteins or F-actin. Scale bar, 10 µm. Ctrl, control cells. The arrows point to the leading edges. (**C**) Data are mean values of experiments from at least 20 cells for each group. Error bars indicate SD. ** *p* < 0.01. One-way ANOVA was used for statistical analysis.

**Figure 6 cells-11-02334-f006:**
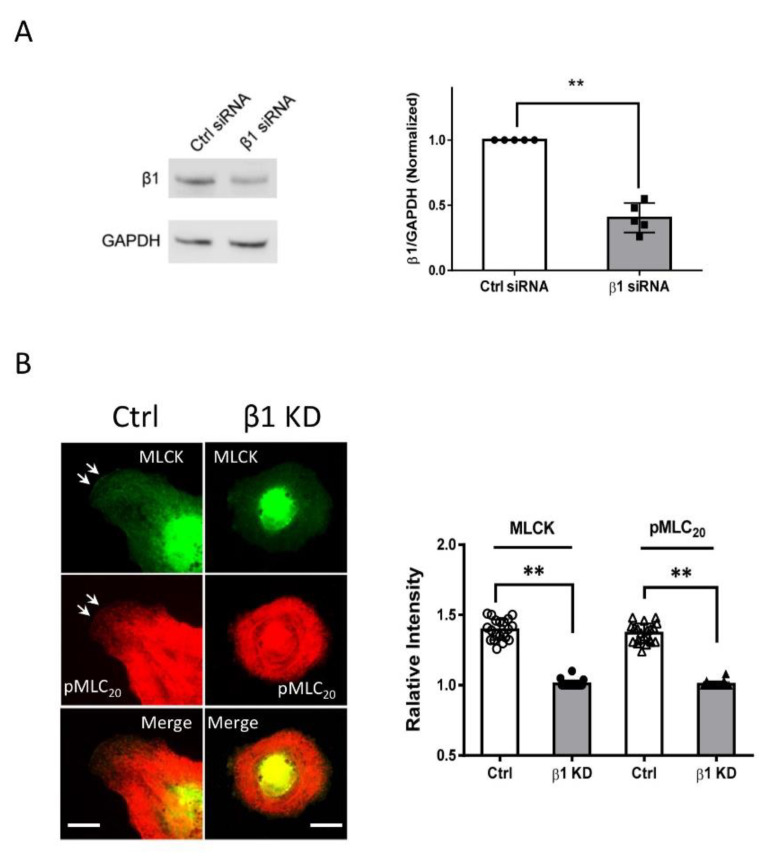
**Integrin β1 controls the positioning of MLCK at the tip of the protrusion.** (**A**) Human airway smooth muscle (HASM) cells treated with control (Ctrl) siRNA or β1 siRNA were evaluated by immunoblot analysis. Data are mean values of experiments from five cultures from three donors. Error bars indicate SD. (**B**) β1 KD attenuates the positioning of MLCK and pMLC_20_ at the cell edge. The arrows point to the leading edge. Scale bar: 10 µm. Data are mean values of experiments from at least 20 cells for each group. Error bars indicate SD. ** *p* < 0.01. The *t*-test was used for statistical analysis.

**Figure 7 cells-11-02334-f007:**
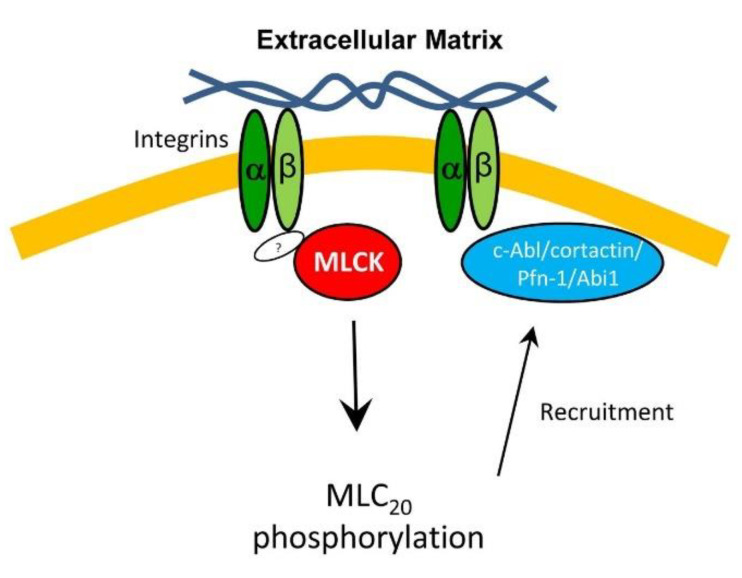
**Proposed mechanism:** In response to extracellular cues (e.g., the ECM), integrin β1 locates at the tip of lamellipodia, which recruits MLCK to the leading cell edge via an unknown mechanism. The kinase then catalyzes MLC_20_ phosphorylation, which promotes the recruitment of c-Abl, cortactin, Pfn-1, and Abi1 to the leading edge, lamellipodial formation, and migration.

## Data Availability

Essential datasets supporting the conclusions are included in this published article.

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
