# Peer review of "Smooth Muscle Myosin Localizes at the Leading Edge and Regulates the Redistribution of Actin-regulatory Proteins during Migration"

_cells, 2022, doi:10.3390/cells11152334_

Round 1

Reviewer 1 Report

The authors found that 20-kDa myosin light chain (MLC20) and 13 myosin-11 (MYH11) localize at the edge of lamellipodia. Knockdown (KD) of these proteins attenuated the recruitment of actin-regulatory proteins such as c-Abl, cortactin, Pfn-1, and Abi1 to the cell edge. Myosin light chain kinase (MLCK), which colocalizes with integrin β1 at the tip of protrusion, is a predominant kinase that catalyzes MLC20 phosphorylation to activate myosin. Inhibition of MLCK attenuated the recruitment of the actin-regulatory proteins to the cell edge. Finally, KD of integrin β1  reduces MLCK localization at leading edge. Therefore, the authors concluded that integrin β1 controls myosin activation and recruitment of actin-regulatory proteins myosin activation to the leading edge. Although the authors claimed that this is the first evidence to suggest that myosin localizes at leading cell edge, non-muscle myosin II A and B are known to localize to the leading edge (Non-muscle myosin II isoforms orchestrate substrate stiffness sensing to promote cancer cell contractility and migration: Cancer Lett, 2022 Jan 1;524:245-258)

Major issue:

Human cells express many kinds of myosin-heavy chains and light chains. The authors need to explain why they focused on MLC20 and MYH11. For example, does human airway smooth muscle (HASM) cells express other heavy and light chains? If so, how much compared to MLC20 and MYH11? Do other heavy and light chains also behave like MLC20 and MYH11?

Minor comment:

Specificity of the antibodies used are key for this project. The authors should provide the evidence that all the antibodies used are specific.

Author Response

We thank the reviewer for the constructive criticisms and suggestions. We have revised the manuscript to address these concerns and suggestions.  Below are detailed our responses:

Comment 1: “Although the authors claimed that this is the first evidence to suggest that myosin localizes at leading cell edge, non-muscle myosin II A and B are known to localize to the leading edge (Non-muscle myosin II isoforms orchestrate substrate stiffness sensing to promote cancer cell contractility and migration: Cancer Lett, 2022 Jan 1;524:245-258)”

Response 1: The reviewer is correct. Non-muscle myosin IIA localize to the leading edge. We have revised the section of Introduction and Discussion. Please see first paragraph of Introduction and Discussion.

Comment 2: “Human cells express many kinds of myosin-heavy chains and light chains. The authors need to explain why they focused on MLC20 and MYH11. For example, does human airway smooth muscle (HASM) cells express other heavy and light chains? If so, how much compared to MLC20 and MYH11? Do other heavy and light chains also behave like MLC20 and MYH11?”

Response 2: Thanks for the excellent question. MYH11 is smooth muscle-specific heavy chain. Based on our Western blots, HASM cells just have MYH11. Other MYH isoforms exist in other cell types. For example, MYH1, MYH2, and MYH4 are fast skeletal muscle specific. MYH3 is for embryonic muscle. MYH6-7 is for cardiac muscle. MYH9, MYH10 and MYH14 are non-muscle myosin. In addition, based on our Western blot analysis, HASM cells just have one isoform of MLC20.      

Comment (Minor) 3: “Specificity of the antibodies used are key for this project. The authors should provide the evidence that all the antibodies used are specific.”

Response 3: Thanks for asking the question. Antibodies against MLC20, integrin β1, MYH11, c-Abl, Abi1, and cortactin were validated by KD experiments. Antibodies against GAPDH, MLCK, and Pfn-1 were validated by examining molecular weight of detected bands. This information is provided in Materials and Methods.

Reviewer 2 Report

It is a very interesting study on an additional role of proteins of the contractile apparatus (MLCK, MLC2, MYH11 etc.) in the mobility of smooth muscle cells and their possible interaction with integrin beta1 and cortical actin.

Minor comments:

1. The citation for line 110 is missing.

2. The use of KD throughout the document has different meanings, either as a knock down or as an antibody affinity. It is recommended that it does not cause confusion, please clearly establish its meaning.

3. It seems that there are fewer cells in the MYH11 KD group, how many cells did they have per well in the healing assay?

4. The HASM abbreviation in figure 1 does not make sense because it is not mentioned in the figure or in the figure legend; however, in other figures it appears without mentioning its meaning. Please put the meaning of the abbreviations of each figure in the figure caption to make it easier to understand.

5. In some figures, such as figure 3, there is no size bar showing the magnification, and it gives the impression that the cells of the control group of this graph are larger than those of the rest of the figures. Please include magnification bars on each figure.

6. The letter D is missing in figure 2, and the letter C in figure 6.

7. In figure 6, the relative intensity of MLCK and MLC20 shown in the figure was obtained from the whole cell or at the cell edge?

8. For the reviewer some conclusions are not clear, in particular it is not convincing to consider whether the beta1 integrin is responsible for the recruitment of MLCK, the reviewer consider that more experiments are required to conclude this role of beta1 integrin. In figure 7 it would be convenient to put something between integrin and MLCK because a direct interaction has not been demonstrated.

Author Response

We thank the reviewer for the constructive criticisms and suggestions. We have revised the manuscript to address these concerns and suggestions.  Below are detailed our responses:

Comment 1: “It is a very interesting study on an additional role of proteins of the contractile apparatus (MLCK, MLC2, MYH11 etc.) in the mobility of smooth muscle cells and their possible interaction with integrin beta1 and cortical actin.”

Response 1: Thanks for the positive comment.

Minor comments:

Comment 2: “The citation for line 110 is missing.”

Response 2: Thanks for pointing out. Have been added.

Comment 3: “The use of KD throughout the document has different meanings, either as a knock down or as an antibody affinity. It is recommended that it does not cause confusion, please clearly establish its meaning.”

Response 3: Thanks for the suggestion. We have a section of “List of Abbreviations” which define “KD” as knockdown in this manuscript.

Comment 4: “It seems that there are fewer cells in the MYH11 KD group, how many cells did they have per well in the healing assay?”

Response 4: We seed 80,000 cells to 6-well plate and grow 2 days until >95% confluence. Typically, less cell density allows quicker migration. But, in this case, MYH11 KD cells have slightly lower density with slower migration. This provides another piece of evidence that MYH11 KD reduces migration.

Comment 5: “The HASM abbreviation in figure 1 does not make sense because it is not mentioned in the figure or in the figure legend; however, in other figures it appears without mentioning its meaning. Please put the meaning of the abbreviations of each figure in the figure caption to make it easier to understand.”

Response 5: Done.

Comment 6: “In some figures, such as figure 3, there is no size bar showing the magnification, and it gives the impression that the cells of the control group of this graph are larger than those of the rest of the figures. Please include magnification bars on each figure.”

Response 6: Cortactin and Pfn-1 images are double staining. Abi1 and F-actin are also double staining. In original figure, we just include scale bar in one of the staining. Agree with the reviewer, it is better to add scale bar to each image.

Comment 7: “The letter D is missing in figure 2, and the letter C in figure 6.”

Response 7: Corrected. Thanks.

Comment 8: “In figure 6, the relative intensity of MLCK and MLC20 shown in the figure was obtained from the whole cell or at the cell edge?”

Response 8: All intensity of staining for this paper were obtained from cell edge.

Comment 9: For the reviewer some conclusions are not clear, in particular it is not convincing to consider whether the beta1 integrin is responsible for the recruitment of MLCK, the reviewer consider that more experiments are required to conclude this role of beta1 integrin. In figure 7 it would be convenient to put something between integrin and MLCK because a direct interaction has not been demonstrated.”

Response 9: Suggestion is accepted.

Reviewer 3 Report

In this study it was found that the regulatory myosin light chain and heavy chain 11 were located at the leading edge of lamellipodia besides the cytoplasm. It was also found that integrin beta1 and MLCK co-localize at the cell edge. The results suggest that phosphorylation of MLC20 by MLCK facilitates recruitment of actin associated proteins c-Abl, cortactin, Pfn-1, and Abi1 to the cell edge and promotes formation of lamellipodia and cell migration. The work constitutes a significant contribution to our knowledge on smooth muscle cell migration.

Major

I believe Reference 37 (Zhang and Gunst, J Physiol 595.13 (2017) pp 4279–4300) need to be discussed in a greater detail, because there is some conceptual overlap in terms of the contraction model and also it raises a question about the identity of myosin found at the cell edge. Non-muscle myosin (NM) has been found in smooth muscle and shown to form filaments in the cell cortex upon phosphorylation (Ref 37). I believe the antibody used in the present study cannot distinguish between SM and NM myosin. If that is the case, then the MLC20 and MYH11 found at the cell edge could belong to NM. A discussion on the present findings in relation to those from Reference 37 would be helpful to your readers, especially in terms of what’s truly novel about the present study. For example, in line 35-36 it is stated that “It is currently unknown whether myosin localizes in other structures of the cell”, this is not true when considering that NM myosin was found (Ref 37) to be located in the cell cortex and its activation is associated with the recruitment of actin binding proteins to the focal adhesion site.

Although this may be out of the scope of this study, but the authors may want to discuss in more details about the sequence of MLC20 phosphorylation in the cytosol and cortex space. One would suspect that MLC20 phosphorylation in the cell cortex should proceed first, ahead of the MLC20 phosphorylation in the cytosolic space. This is because the cellular force transmission structures should be assembled before force is generated by the stress fibers. I don’t think this is known, so any discussion will be speculative in nature.  

Minor

Line 36: “because study on this”. Suggested change: “because studies on intracellular locations of myosin”.

Line 41: “decide moving direction”. Suggested change: “decide on a moving direction”.

Line 42: “at front”. Suggested change: “at the leading cell edge along the migration path”.

Line 53: “Arp2/3”. The abbreviation needs to be spelled out in its first appearance.

Line 58: “affect”, should be “affects”.

There are more minor typos throughout the manuscript, the authors need to go over the text carefully for the final version.

Line 312-313: “the shortening of myosin filaments can break actin filaments”. I don’t think a myosin filament can shorten itself. A more correct description is needed here.

Line 314: “for next round polymerization”. Suggested change: “for the next round of polymerization”.

Author Response

We thank the reviewer for the constructive criticisms and suggestions. We have revised the manuscript to address these concerns and suggestions.  Below are detailed our responses:

Comment 1: “In this study it was found that the regulatory myosin light chain and heavy chain 11 were located at the leading edge of lamellipodia besides the cytoplasm. It was also found that integrin beta1 and MLCK co-localize at the cell edge. The results suggest that phosphorylation of MLC20 by MLCK facilitates recruitment of actin associated proteins c-Abl, cortactin, Pfn-1, and Abi1 to the cell edge and promotes formation of lamellipodia and cell migration. The work constitutes a significant contribution to our knowledge on smooth muscle cell migration.”

Response 1: Thanks for the positive comment.

Comment 2: “I believe Reference 37 (Zhang and Gunst, J Physiol 595.13 (2017) pp 4279–4300) need to be discussed in a greater detail, because there is some conceptual overlap in terms of the contraction model and also it raises a question about the identity of myosin found at the cell edge. Non-muscle myosin (NM) has been found in smooth muscle and shown to form filaments in the cell cortex upon phosphorylation (Ref 37). I believe the antibody used in the present study cannot distinguish between SM and NM myosin. If that is the case, then the MLC20 and MYH11 found at the cell edge could belong to NM. A discussion on the present findings in relation to those from Reference 37 would be helpful to your readers, especially in terms of what’s truly novel about the present study. For example, in line 35-36 it is stated that “It is currently unknown whether myosin localizes in other structures of the cell”, this is not true when considering that NM myosin was found (Ref 37) to be located in the cell cortex and its activation is associated with the recruitment of actin binding proteins to the focal adhesion site.”

Response 2: Thanks for the excellent question. We have change “myosin” to “smooth muscle myosin” (Please see the section of Introduction). We have also discussed more about Reference 39 (in current version) in detail (in Discussion). In this study, we used MYH11 antibody, which selectively recognizes smooth muscle myosin. This is verified by CRISPR/Cap knockdown experiment. NMIIA is MYH9, NMIIB is MYH10. They are detected by different antibodies.

Comment 3: “Although this may be out of the scope of this study, but the authors may want to discuss in more details about the sequence of MLC20 phosphorylation in the cytosol and cortex space. One would suspect that MLC20 phosphorylation in the cell cortex should proceed first, ahead of the MLC20 phosphorylation in the cytosolic space. This is because the cellular force transmission structures should be assembled before force is generated by the stress fibers. I don’t think this is known, so any discussion will be speculative in nature.”

Response 3: This is an excellent idea. We have added this speculation in Discussion.

Comment 4: Line 36: “because study on this”. Suggested change: “because studies on intracellular locations of myosin”.

Response 4: Thanks. Have changed as suggested.

Comment 5: Line 41: “decide moving direction”. Suggested change: “decide on a moving direction”.

Response 5: Changed as suggested.

Comment 6: Line 42: “at front”. Suggested change: “at the leading cell edge along the migration path”.

Response 6: Changed as suggested.

Comment 7: Line 53: “Arp2/3”. The abbreviation needs to be spelled out in its first appearance.

Response 7: Done.

Comment 8: Line 58: “affect”, should be “affects”.

Response 8: Done.

Comment 9: There are more minor typos throughout the manuscript, the authors need to go over the text carefully for the final version.

Response 9: We have double checked for typos.

Comment 10: Line 312-313: “the shortening of myosin filaments can break actin filaments”. I don’t think a myosin filament can shorten itself. A more correct description is needed here.

Response10: Changed to “sliding of contractile filaments, which can break actin filaments”

Comment 11: Line 314: “for next round polymerization”. Suggested change: “for the next round of polymerization”.

Response 11: Done.

Reviewer 4 Report

In this manuscript, Wang and colleagues provide a compelling evidence that myosin-11 and its core regulatory and activated component (MLC20-P) are localized to the cellular periphery of lamellipodia.  The authors showed new mechanistic data suggesting that these subcellular localization of myosin is regulated by beta1 integrin and its subsequent recruitment of MLCK to the leading edge.  Importantly, the authors showed that the function of MLCK-activated MLC20 phosphorylation at the lamellipodia is to recruit actin-regulatory proteins that facilitate cellular migration in F-actin dependent manner.  The rationale for the study is well-conceived, experiments are well-designed, and interpretation of the results are sound, offering new molecular understanding of airway smooth muscle migration.  These results would provide further investigations into regulation of these events in disease such as asthma.

Few minor comments:

Abstract:

Please change "repairing" to "repair"

Start with "Traditionally, smooth muscle myosin II has been thought to ..."

Delete or incorporate these two statements with the results/findings. "Morevover, myosin light chain kinase is a predominat kinase..." and "Integrin b1 is critical for migration of various cell types"

Introduction:

Line 29: two regulatory 20kDa myosin light chain (MLC20) and two essential 17kDa myosin light chains (MLC17).

Line 54: ...the mechanisms that regulate lamellipodial formation remain unclear.

Materials and Methods:

Line 90: please define the size of the wells used.

Results:

Line 172: Please change "directional" to "cellular" migration.

Does inhibition of any of the actin-regulatory proteins recruitment sufficient for inhibition of cellular migration?

Does the spatial distribution of myosin and its recruitment of actin-regulatory proteins change during cellular migration? 

Author Response

We thank the reviewer for the constructive criticisms and suggestions. We have revised the manuscript to address these concerns and suggestions.  Below are detailed our responses:

Comment 1: “In this manuscript, Wang and colleagues provide a compelling evidence that myosin-11 and its core regulatory and activated component (MLC20-P) are localized to the cellular periphery of lamellipodia.  The authors showed new mechanistic data suggesting that these subcellular localization of myosin is regulated by beta1 integrin and its subsequent recruitment of MLCK to the leading edge.  Importantly, the authors showed that the function of MLCK-activated MLC20 phosphorylation at the lamellipodia is to recruit actin-regulatory proteins that facilitate cellular migration in F-actin dependent manner.  The rationale for the study is well-conceived, experiments are well-designed, and interpretation of the results are sound, offering new molecular understanding of airway smooth muscle migration.  These results would provide further investigations into regulation of these events in disease such as asthma.”

Response 1: Thanks for the positive comment.

Comment 2: “Abstract: Please change "repairing" to "repair"”

Response 2: Thanks for the suggestion. Done.

Comment 3: “Start with "Traditionally, smooth muscle myosin II has been thought to ..." Delete or incorporate these two statements with the results/findings. "Moreover, myosin light chain kinase is a predominat kinase..." and "Integrin b1 is critical for migration of various cell types"

Response 3: We thought that the journal has a broad spectrum of readers. Some of them may not know what MLCK and Integrin beta1 are. So, we provided a short background. However, in response to this reviewer, we have deleted the two sentences. 

Comment 4: “Line 29: two regulatory 20kDa myosin light chain (MLC20) and two essential 17kDa myosin light chains (MLC17). Line 54: ...the mechanisms that regulate lamellipodial formation remain unclear.”

Response 4: Done.

Comment 5: “Materials and Methods: Line 90: please define the size of the wells used.”

Response 5: They are 6-well plate. Have been added.

Comment 6: “Results: Line 172: Please change "directional" to "cellular" migration. Does inhibition of any of the actin-regulatory proteins recruitment sufficient for inhibition of cellular migration? Does the spatial distribution of myosin and its recruitment of actin-regulatory proteins change during cellular migration?”

Response 6: Thanks for the excellent questions. 1. Have changed to “cellular.”  2. Previous studies by our group and others have shown that disruption of the recruitment of actin-regulatory proteins inhibits migration. For example, an Abi1 mutant not localized at leading edge inhibits migration. We have added this to Discussion. 3. In this study, we assessed the spatial distribution of myosin and the actin-regulatory proteins in cells after plating them on collagen-coated coverslips. This is a standard way to determine spatial localization of protein during migration. After plating, cells form lamellipodia and migration. In addition, previous studies by our laboratory and others have shown that labeled actin-regulatory proteins localized at the leading edge in live cells during migration. We have added this in Discussion (See 2nd paragraph of Discussion).     

Round 2

Reviewer 1 Report

The authors have satisfactorly answered to all my questions.